# Gene Suppression Therapies in Hereditary Cerebellar Ataxias: A Systematic Review of Animal Studies

**DOI:** 10.3390/cells12071037

**Published:** 2023-03-29

**Authors:** Carolina Santos, Sofia Malheiro, Manuel Correia, Joana Damásio

**Affiliations:** 1Infectious Diseases Department, Centro Hospitalar Lisboa Ocidental, 1449-005 Lisboa, Portugal; 2Neurology Department, Centro Hospitalar Universitário do Porto, 4099-001 Porto, Portugal; 3ICBAS School of Medicine and Biomedical Sciences, Universidade do Porto, 4050-318 Porto, Portugal; 4CGPP, IBMC-Institute for Molecular and Cell Biology, i3S-Instituto de Investigação e Inovação em Saúde, Universidade do Porto, 4200-135 Porto, Portugal; 5UnIGENe, IBMC-Institute for Molecular and Cell Biology, i3S-Instituto de Investigação e Inovação em Saúde, Universidade do Porto, 4200-135 Porto, Portugal

**Keywords:** hereditary spinocerebellar degenerations, Machado–Joseph disease, gene silencing, RNA interference

## Abstract

Introduction: Hereditary cerebellar ataxias (HCAs) are a heterogenous group of neurodegenerative disorders associated with severe disability. Treatment options are limited and overall restricted to symptomatic approaches, leading to poor prognoses. In recent years, there has been extensive research on gene suppression therapies (GSTs) as a new hope for disease-modifying strategies. In this article, we aim to perform a review of *in vivo* studies investigating the efficacy and safety profile of GSTs in HCAs. Methods: A structured PubMed^®^ search on GSTs in HCAs from January 1993 up to October 2020 was performed. Inclusion and exclusion criteria were defined, and the selection process was conducted accordingly. The screening process was independently carried out by two authors and was initially based on title and abstract, followed by full-text reading. The risk-of-bias assessment was performed with SYRCLE’s tool. A data extraction sheet was created to collect relevant information from each selected article. Results: The initial search yielded 262 papers, of which 239 were excluded. An additional article was obtained following reference scrutiny, resulting in a total of 24 articles for final analysis. Most studies were not clear on the tools used to assess bias. In SCA1, SCA2, MJD/SCA3 and SCA7, RNA interference (iRNA) and antisense oligonucleotide (ASO) therapies proved to be well tolerated and effective in suppressing mutant proteins, improving neuropathological features and the motor phenotype. In SCA6, the phenotype was improved, but no investigation of adverse effects was performed. In FRDA, only the suppression efficacy of the electroporation of the clustered regularly interspaced short palindromic repeats associated with Cas9 enzyme system (CRISPR-Cas9) system was tested and confirmed. Conclusion: The literature reviewed suggests that GSTs are well tolerated and effective in suppressing the targeted proteins, improving neuropathological features and the motor phenotype *in vivo*. Nonetheless, there is no guarantee that these results are free of bias. Moreover, further investigation is still needed to clarify the GST effect on HCAs such as FRDA, SCA6 and SCA2.

## 1. Introduction

Hereditary cerebellar ataxias (HCAs) are a heterogeneous group of neurodegenerative disorders that mainly affect the cerebellum and its afferent and efferent pathways [1,2]. The majority of dominant HCAs are polyglutamine disorders arising from repeat-expansion triplets (SCA 1, SCA2, MJD/SCA3, SCA6 and SCA7), with the expansion size being partially responsible for age of onset, severity and clinical progression [2,3]. Friedreich ataxia, the most frequent recessive HCA, is caused by a biallelic intronic GAA expansion in the *FXN* gene, and ataxia–telangiectasia (the second most common recessive HCA in some series) arises from point mutations in the *ATM* gene [2,3]. The wild-type (WT) protein role is not completely understood, but several lines of evidence suggest that ataxin-1 (SCA1) regulates gene transcription and ataxin-2 (SCA2) is involved in RNA repair and ribosomal translation, while ataxin-3 (MJD/SCA3) has deubiquitinating activity [2]. The α 1A-Subunit of the voltage-dependent calcium channel of P/Q type (SCA6) regulates neuronal excitability, and ataxin-7 (SCA7) acts as a subunit of histone acetyltransferase complexes [2]. The intronic expansion in *FXN* (Friedreich ataxia), results in the decreased expression of frataxin, a protein involved in mitochondrial iron metabolism, and the ATM protein (ataxia–telangiectasia) is a powerful protein cinase, involved in the cellular response to genotoxic stress [4,5].

Disease-modifying strategies are only available for a minority of hereditary ataxias (e.g., ataxia with vitamin E deficiency), with symptomatic treatment being the only option for the vast majority of patients [6]. The prognosis is generally poor, leading to severe disability and premature death [7].

Gene suppression therapies (GSTs) have been showing relevant progress in the treatment of neurodegenerative disorders (e.g., hereditary transthyretin amyloidosis) [8,9]. GSTs act by lowering the expression of specific genes, and when targeting genes responsible for mutant proteins, reducing mutant protein levels [9]. Of the different types of GSTs, this manuscript will approach RNA interference (iRNA), antisense oligonucleotides (ASOs) and the clustered regularly interspaced short palindromic repeats associated with Cas 9 enzyme system (CRISPR/Cas9 system) [9]. iRNA consists of a double-stranded RNA intracellularly processed by Dicer (RNase III family ribonuclease), which transforms it into small interfering RNA (siRNA). This siRNA is incorporated into the RNA-induced silencing complex (RISC), which, in turn, recognizes the targeted mRNA and destroys it; further, siRNA can lead to translation inhibition [10,11]. ASOs are single-stranded synthetic antisense oligonucleotides that complement a specific mRNA sequence. When binding the targeted mRNA, a variety of mechanisms to attain blocked gene expression may be possible, with mRNA cleavage caused by the induction of RNAse H endo-nuclease being the most common [11,12]. The CRISPR/CAS9 system involves two components and three steps: a guide RNA (gRNA), which recognizes the targeted DNA sequence, and a Cas9 nuclease that cuts DNA and allows it to be repaired by cellular mechanisms [13,14,15].

With this work, we aimed to perform a systematic review of studies investigating the efficacy and safety of GSTs in HCAs and analyse the risk of bias of the included studies.

## 2. Materials and Methods

A protocol was elaborated according to PRISMA guidelines and registered at OSF (registration link: https://osf.io/mpuq7/?view_only=803535457e0e406ead70054efc1071df, accessed on 14 February 2023). The eligibility criteria included studies assessing GST effects on HCAs in animal models or humans. Articles on GSTs in disorders other than HCAs, other therapies in HCAs, review papers and studies reported in languages other than English or Portuguese were excluded. In vitro-only studies were excluded, as the risk of bias is not applicable to this type of studies. In addition, as many had been conducted prior to *in vivo* studies by the same team, we aimed to reduce data duplication.

A structured PubMed search from January 1993 (year of the first description of the underlying genetic defect in HCAs) up to October 2020 was performed. The search strategy included the following MeSH terms: “Hereditary Spinocerebellar Degenerations”, “Friedreich Ataxia”, “Olivopontocerebellar Atrophy”, “Spinocerebellar Ataxia”, “Machado Joseph Disease”, “Gene Silencing” and “RNA Interference”; and the following all fields terms: “Gene Suppression”, “Genome editing”, “Catalytic Nucleic Acids”, “Antisense Oligonucleotides” and “CRISPr-Cas 9 System”. The search was performed as follows: [“Hereditary Spinocerebellar Degenerations” OR “Friedreich Ataxia” OR “Olivopontocerebellar Atrophy” OR “Spinocerebellar Ataxia” OR “Machado Joseph Disease”] AND [“Gene Silencing” OR “RNA Interference” OR “Gene Suppression” OR “Genome editing” OR “Catalytic Nucleic Acids” OR “Antisense Oligonucleotides” OR “CRISPr-Cas 9 System”].

Peer-reviewed articles in English or Portuguese were included, and references were checked to guarantee maximal coverage. The screening process was independently carried out by two authors (C.S. and S.F.), and when divergences were identified, a third author (J.D.) was consulted. All articles were screened by heading and abstract, followed by full-text reading (performed by C.S. and S.F.). Risk-of-bias assessment was conducted using SYRCLE’s risk-of-bias tool, an adapted version of the Cochrane risk-of-bias tool for animal studies [16]. A standardized data sheet was created for data extraction (Appendix A).

## 3. Results

### 3.1. Search Outcome

The PubMed search generated 262 articles, without duplicates. The initial screening, based on headings and abstracts, excluded 196 articles. After full-text reading, 43 additional articles were excluded, and 1 additional article was included. The number of selected studies for analysis was 24. All studies were on animal models, with no study on humans having been identified. A schematic representation of the search results is depicted in Figure 1. The summarized data extraction sheets of the results for each HCAs are reported in Table 1, Table 2, Table 3 and Table 4.

Regarding the risk of bias, most studies were not clear on the categories analysed with the risk-of-bias tool recommended for animal studies (Figure 2). Information on the generation of allocation sequencing and the blinding of caregivers/investigators was lacking in the large majority of the selected studies. No study where both procedures had been performed could be identified.

The results were divided into four main categories: suppression efficacy, neuropathology, motor behaviour and safety profile. For each category, the selected papers used the same outcome measures, with only a few exceptions (detailed above). Suppression efficacy was evaluated according to mutant mRNA and protein levels; effects on neuropathology were assessed using the cerebellar molecular and/or granular layer width, Purkinje cell count, mutant protein inclusion/aggregate count and neuron expression marker recovery. Motor behaviour included tests such as beam waling, footprint pattern analysis and open-field activity. The safety profile was assessed with microglial and astrocytic activation as well as loss of neuron marker expression.

The included studies used different animal models, accommodation conditions and GST dosing, which might affect comparisons and conclusions. Thus, in this result section, we analyse and summarise independent conclusions from the selected papers.

### 3.2. Machado–Joseph Disease

Eleven articles on Machado–Joseph disease (MJD)/spinocerebellar ataxia type 3 (SCA3) were included: nine on iRNA and two on ASOs (Table 1) [17,18,19,20,21,22,23,24,25,26,27].

**Table 1 cells-12-01037-t001:** Key conclusions from data extraction sheet—MJD/SCA3 [17,18,19,20,21,22,23,24,25,26,27].

Author	Year	GST	Key Conclusions
Alves, S., et al.	2008	iRNA	The allele-specific silencing of mutant ataxin-3 was effective and selective *in vivo* and decreased the MJD-associated neuropathological phenotype.
Alves, S., et al.	2010	iRNA	WT ataxin-3 did not reduce the toxicity of mutant ataxin-3; WT overexpression did not protect against MJD neuropathology, and the knockdown of WT did not affect MJD neuropathology. The non-allele-specific silencing of ataxin-3 reduced neuropathology.
Rodríguez-Lebrón, E., et al.	2012	iRNA	iRNA was effective in suppressing *ATXN3*. Administration in a pre-symptomatic mouse model prevented the development of the neuropathological features and motor impairments found in the control group.
Costa, Mdo C., et al.	2013	iRNA	Despite the fact that iRNA was effective in suppressing *ATXN3*, at the end of the study, at 48 weeks of age, no improvement in motor impairment was detected; the authors suggested that the motor phenotype might not be solely due to cerebellar dysfunction or intervention was performed later than ideal. No adverse effects were detected. No differences in lifespan were detected between groups.
Nóbrega, C., et al.	2013	iRNA	iRNA proved to be effective in suppressing *ATXN3*. Its administration after symptom onset prevented the development of MJD-associated motor-behaviour and neuropathological abnormalities.
Nóbrega, C., et al.	2014	iRNA	The effective gene silencing of *ATXN3* in pre-symptomatic mice led to the clearance of mutant ataxin-3 from neuronal nuclei and prevented the development of motor impairments. There were no differences between groups concerning glial or astrocytic activation.
Conceição, S., et al.	2016	iRNA	Intravenous administration was successful in crossing the BBB. iRNA was effective in mutant ataxin-3 knockdown *in vivo*. iRNA improved motor performance and recovered striatal- and cerebellar-associated neuropathology. No signs of toxicity were detected.
Li, Y.X., et al.	2018	iRNA	The downregulation of Relish expression in astrocytes delayed neurodegeneration and extended the lifespan in the SCA3 fly model.
Nóbrega, C., et al.	2018	iRNA	While evidence of neuronal dysfunction and gliosis was present at initial timepoints, 20 weeks post-injection, no differences between groups were found. No off-target effects or saturation of the endogenous iRNA processing machinery in the mouse striatum were detected.
Evers, M.M., et al.	2013	ASOs	The intracerebral injection of ASOs was effective in skipping targeted exons. No overt toxicity was observed *in vivo*.
McLoughlin, H.S., et al.	2018	ASOs	ASOs achieved the efficient silencing of mutant *ATXN3* and prevented the nuclear accumulation of ataxin-3 protein. Administration in post-symptomatic mice fully recovered locomotor activity. No signs of an adverse immune response to treatment were detected.

#### 3.2.1. Suppression Efficacy

Suppression efficacy was evaluated in ten articles: eight on iRNA and two on ASOs [17,18,19,20,21,22,23,24,26,27]. Seven studies identified a significant decrease in mutant ataxin-3 mRNA and/or protein levels compared with the control group [18,19,20,23,25,26,27]. The extent of the reduction in mutant transcript levels varied from 32% to 92%, and both extremes were attained with iRNA [19,25]. Regarding mutant protein levels, the values varied from 32% to 91%, with the highest reduction having been reached with ASOs [19,23].

Alves et al. confirmed iRNA suppression efficacy by verifying a decrease in the expression of mutant ataxin-3 using immunohistochemistry [17]. Costa et al. analysed mutant ataxin-3 levels 9–10 months after iRNA injection (end-of-life stage) and identified a reduction of up to 40% in treated mice, suggesting effectiveness until end-stage disease [20]. Evers et al. proved that an ASO targeting the exons responsible for CAG repeat expansion was effective in reducing targeted exon levels [21]. Li et al. assessed whether the knockdown of relish, an astrocyte-specific NF-kB transcription factor, or relish-dependent AMPs could attenuate eye degeneration in MJD/SCA3 Drosophila models. Indeed, knockdown efficiency was confirmed by assessing relish expression and relish-dependent AMP levels [22].

#### 3.2.2. Neuropathology

Effects on neuropathology were studied in eight publications: seven on iRNA and one on ASOs [17,18,19,22,23,25,26,27]. Neuronal mutant ataxin-3 inclusions were assessed in all studies, with all, except for Li et al.’s, demonstrating a decrease in neuronal inclusion. Alves et al., comparing anti-ataxin-3 iRNA- and vehicle-treated mice, identified relative decreases of 48,2% in the number and 12,7% in the size of mutant ataxin-3-positive inclusions [17]. McLoughlin and collaborators proved ASO treatment to be effective in reducing neuronal nuclear ataxin-3 accumulation compared with vehicle-treated WT until 22 weeks of age. The group receiving control ASOs not targeting mutant ataxin-3 also presented a significant reduction in neuronal and non-neuronal ataxin-3 nuclear accumulation when compared with vehicle-injected mice [23]. The authors suggested that this could be a result of a non-specific effect of ASOs requiring further investigation [23].

Neuronal loss and dysfunction analysis was evaluated in six articles, demonstrating significant neuronal preservation in the intervention group [17,18,19,23,25,26]. DARPP-32 immunoreactivity analysis was performed in five papers on iRNA, with all papers identifying higher DARPP-32 marker expression in the treated group [17,18,19,25,26]. DARPP-32 is a striatal neuronal marker whose expression is lost when neuronal dysfunction is present. Alves et al. revealed the greatest DARPP-32 immunoreactivity recovery (~70%) [17]. The same authors also verified a marked decrease in the number of degenerating neurons and atrophic nuclei in iRNA-treated mice and no signs of striatal atrophy compared with controls [17]. Of the two works that analysed the Purkinje cell count, only one verified significant preservation in the iRNA-treated group, i.e., 11.85 vs. 7 in the control [19,25]. McLoughlin et al. evaluated Purkinje cell function by assessing the firing frequency and identified the recovery of this phenotype in the intervention group [23].

The cerebellar granular layer width was evaluated in three articles, and all identified significant atrophy attenuation in iRNA-treated mice [19,25,26]. Likewise, the two papers studying the molecular cerebellar layer detected a significantly thicker layer in iRNA-treated mice [25,26]. Nóbrega et al. identified 145.1 µm molecular layer width in the intervention group and 116.3 µm width in controls, as well as dendritic arborization preservation [25,26].

#### 3.2.3. Motor Behaviour

Motor behaviour was assessed in five studies: four on iRNA and one on ASOs [19,20,23,25,26]. All disclosed significant improvement in motor behaviour when compared with untreated models and/or similar performances when compared with WT.

Regarding therapy administration timing, one study found that the pre-symptomatic administration of iRNA prevented the deterioration of balance, motor coordination, gait and hyperactivity [25]. McLoughlin et al. showed that ASO administration in early symptomatic mice fully recovered locomotor activity until 29 weeks of age [23]. In another study, iRNA administration in mice with severe symptoms improved motor performance; nonetheless, it was still far from the performance of WT [26]. Costa et al. demonstrated that despite the effective lifelong suppression of mutant ataxin-3 and effective delivery to the cerebellum, iRNA administered to early symptomatic mice failed to improve motor impairment during the whole duration of the study (48 weeks) [20]. The authors suggested that the motor phenotype might not be solely due to cerebellar dysfunction, with other CNS regions also being involved [20]. The timing of treatment administration could also be relevant, with later administration probably not being sufficient to revert symptoms that are already present [19].

Four studies confirmed that intrathecal injection attained effective cerebellar distribution of the GST and improved motor behaviour [20,23,25,26], while one study revealed that iRNA peripheral injection also resulted in improved motor behaviour [19].

#### 3.2.4. Wild-Type Protein

Two articles on iRNA focused on the role of WT ataxin-3 [17,18]. Alves et al. demonstrated that the allele-specific silencing of mutant ataxin-3, while keeping WT levels, led to the attenuation of the neuropathological phenotype [17]. In another work, the authors concluded that the overexpression of WT ataxin-3 did not protect against the MJD/SCA3 neuropathological phenotype and was associated with increased toxicity [18]. When assessing the effects of knocking down ataxin-3 in WT rats, the authors found no differences in neuropathological features between WT rats with lower levels of ataxin-3 protein and the control group [18].

#### 3.2.5. Safety Profile

The safety profile was evaluated in five studies (four on iRNA and one on ASOs) [20,22,23,24,27], and lifespan, in two [20,22]. Costa et al. did not find significant differences in lifespan between iRNA-treated and control-treated mice, detecting a similar average age of death at 58 weeks [20]. In contrast, Li et al. found extended lifespan in iRNA-treated flies [22].

Two studies evaluated neurotoxicity associated with iRNA injection by assessing DARPP-32 immunoreactivity and/or NeuN expression [20,24]. Ten weeks post-injection, Costa et al. found no differences in NeuN-positive cell distribution and density between treated and untreated mice [20]. Nóbrega et al. concluded that surgical injection led to the loss of DARPP-32 and NeuN expression 2 weeks post-treatment and recovery at 8 weeks. In fact, neuronal recovery was more pronounced in iRNA-treated subjects than in controls [24].

Four articles assessed the effects on glial activation—Gfap marker—and microglial activation—Iba1 marker [20,23,24,27]. None found differences in glial activation markers between intervention and controls. Nevertheless, Nóbrega et al. detected glial activation 2 weeks post-injection, either with a control vector or a vector containing iRNA [24]. However, 8 and 20 weeks post-injection, no differences in Gfap and Iba1 markers were found between the injected and non-injected groups, indicating a transient inflammatory response [24].

Nóbrega et al. also studied the iRNA administration off-target effects by measuring the levels of endogenous transcripts with sequences pairing the seed region of iRNA [24]. It was shown that 2 and 20 weeks post-injection, the quantitative analysis revealed no differences between injected and non-injected mice [24]. In the same study, the saturation of endogenous iRNA processing machinery, with Drosha, Dicer and exportin-5 mRNA levels, was analysed, and no significant differences between injected and non-injected mice were identified, discarding iRNA pathway saturation [24].

Finally, to assess cellular toxicity due to increasing doses of ASOs, McLoughin et al. evaluated the ratio of pro- and anti-apoptotic proteins (BAX/BCL2) [23]. Four weeks post-injection, there was no increase in BAX/BCL2 with the increase in ASO doses in the intervention group relative to WT and controls. The authors concluded there was no cellular toxicity at the tested doses [23].

### 3.3. Spinocerebellar Ataxia Type 1

Six papers on SCA1 and GSTs were included: five on iRNA and one on ASOs (Table 2) [28,29,30,31,32,33].

**Table 2 cells-12-01037-t002:** Key conclusions from data extraction sheet—SCA1 [28,29,30,31,32,33].

Authors	Year	GST	Key Conclusions
Xia, H., et al.	2004	iRNA	iRNA reduced ataxin-1 transcript levels, resulting in improved motor coordination, restored cerebellar morphology and resolved characteristic ataxin-1 inclusions in the Purkinje cells of SCA1 mice.
Keiser, M.S., et al.	2013	iRNA	The silencing of mutant ataxin-1 using miRNAs recovered behavioural deficits and improved neuropathology. It is suggested that behavioural recovery does not require the full recovery of all neuropathological aspects.
Keiser, M.S., et al.	2014	iRNA	Reduced ataxin-1 transcript and protein levels without overt neurotoxicity were achieved. It preserved cerebellar lobule integrity for over a year and preserved rotarod performance for months (30 w).
Keiser, M.S., et al.	2015	iRNA	Anti-ataxin-1 iRNA resulted in the reduction in *ATXN1* mRNA, and no signs of toxicity were detected.
Keiser, M.S., et al.	2016	iRNA	The iRNA-mediated suppression of ataxin-1 mRNA altered disease progression, reversed motor symptoms and normalized cerebellar pathology when delivered before and after symptom onset.
Friedrich, J., et al.	2018	ASOs	Following a single ASO injection at 5 weeks of age, ataxin-1 transcripts remained reduced until 18 w, but ataxin-1 protein only remained at reduced levels in pons at 18 w. Nonetheless, mice demonstrated recover of neuropathological and motor behaviour phenotypes.

#### 3.3.1. Suppression Efficacy

Suppression efficacy was assessed in all studies, and overall, the tested therapies were effective in reducing ataxin-1 mRNA and/or protein levels [28,29,30,31,32,33]. Keiser et al. showed that iRNA led to the sustained suppression of ataxin-1 transcripts; in particular, three weeks after injection, iRNA-treated mice had 30% of the total ataxin-1 transcripts present in control-treated mice, and 75%, 35 weeks after injection [30]. Friedreich et al. demonstrated that a single ASO injection reduced ataxin-1 transcript levels from two to eighteen weeks post-injection [28]. However, ataxin-1 levels eighteen weeks after injection were not homogenous in the CNS; there was a significant reduction in the pons, whereas in the medulla, cerebral cortex and cerebellum, it returned to untreated levels [28].

#### 3.3.2. Neuropathology

Neuropathology was assessed in four articles [29,30,32,33]. Concerning cerebellar molecular layer width, all studies demonstrated that iRNA anti-ataxin-1-treated mice had a thicker molecular layer than the saline-treated group, and no significant differences compared to WT mice were noted. Xia et al. found the widths of 162 μm in the intervention group and 158 μm in the WT group [33]. Keiser et al. demonstrated that iRNA recovered this neuropathological feature when therapy was administered either before or after symptom onset [32].

Xia et al. also studied the effect of iRNA anti-ataxin-1 on intraneuronal inclusions. iRNA-treated mice presented complete resolution of inclusions in transduced Purkinje cells [33]. Keiser et al. found significant recover of Purkinje cell number, as well as a reduction in ectopic cells [30].

#### 3.3.3. Motor Behaviour

Motor behaviour was assessed in five studies, and overall, treated groups performed better than controls [21,29,30,32,33]. Keiser et al. found that iRNA-treated mice showed sustained, longer strides and wider hindlimb stances than controls at 30 and 40 weeks of age [29]. In addition, regarding administration timing, the same authors reported that iRNA administration, before or after symptom onset, led to the improvement in rotarod test performance compared with control or untreated mice [32]. Nonetheless, when administered in the pre-symptomatic stage, iRNA-treated animals’ performance was similar to that of WT littermates, in contrast to administration after symptom onset, which resulted in a performance still far from that of WT [32]. Friedreich et al. described ASO injection in a very early stage of disease to improve performance in the rotarod test and beam walking analysis, whereas administration in an early–mid-stage of disease only led to significant differences in beam walk and not in the rotarod test [28].

#### 3.3.4. Safety Profile

Safety was evaluated by assessing microglial and astrocytic activation marker (Iba1 and Gfap, respectively) expression in four articles, all by Keiser and collaborators [29,30,31,32]. No differences in Iba1 or Gfap expression at the injection site or cerebellar lobules between iRNA anti-ataxin-1- and saline-treated mice were identified [29,30]. In another study, a slight enhancement of both markers in the injected cerebellar cortex hemisphere, when compared with the untreated hemisphere, was identified [31]. Keiser et al. also reported a slight enhancement of the astrocytic marker at the injection site in all injected mice and no differences in microglial activation marker expression when compared with the control group [32]. Nonetheless, all concluded that GSTs were well tolerated.

### 3.4. Spinocerebellar Ataxia Type 7

Three articles on spinocerebellar ataxia type 7 (SCA7) and GSTs were selected: two on iRNA and one on ASOs [34,35,36]. Two of these studies analysed retinal degeneration with GSTs being administered with subretinal injection or injection into vitreous body (Table 3) [34,35].

**Table 3 cells-12-01037-t003:** Key conclusions from data extraction sheet—SCA7 [34,35].

Authors	Year	GST	Key Conclusions
Ramachandran, P.A.S., et al.	2014	iRNA	A sustained reduction in ataxin-7 expression led to significant and robust improvements in the ataxic and neuropathological phenotypes as well as delayed disease onset in SCA7 mice. No significant adverse effects were present.
Ramachandran, P.A.S., et al.	2014	iRNA	Preservation of normal retinal function 23 weeks post-retinal injection and no adverse toxicity with reduction in ataxin-7 transcript levels were reported.
Niu, C., et al.	2018	ASOs	ASOs were effective in suppressing mutant ataxin-7 transcript and protein levels; visual function was improved despite initiating treatment after symptom onset. At the end of the study, ataxin-7 ASOs only ameliorated rod photoreceptor function. CAG repeat-targeting ASOs were less effective than Ataxin-7 ASOs.

#### 3.4.1. Suppression Efficacy

All three articles assessed suppression efficacy and identified decreased mutant ataxin-7 transcript and protein levels when compared with placebo or untreated mice. Niu et al. showed a >60% reduction in ataxin-7 mRNA levels 6 weeks after ASO retinal injection [34]. Similarly, Ramachandran et al. identified sustained reductions of 50% in ataxin-7 mRNA levels and of 35% in ataxin-7 protein levels 33 weeks after intrathecal iRNA injection [36].

#### 3.4.2. Neuropathology

Neuropathology was assessed in two studies [34,36]. Niu et al. observed that ASO anti-ataxin-7 administration at the time of or after symptom onset resulted in a significant reduction in intraneuronal inclusions [34]. This same paper concluded that ASOs targeting the CAG repeat had a less clear reduction in ataxin-7 aggregates than anti-ataxin-7 ASOs. Ramachandran et al. demonstrated that anti-ataxin-7 iRNA intrathecal injection resulted in an approximate 80% reduction in neuronal inclusions in transduced Purkinje cells when compared with controls [36]. The molecular cerebellar layer width was assessed by Ramachandran et al., who identified an increase of 10% in anti-ataxin-7 iRNA-treated mice relatively to controls [36].

#### 3.4.3. Motor Behaviour

Motor behaviour was only assessed in one study [36]. Anti-ataxin-7 iRNA-treated mice presented a significant improvement in hindlimb clasping score, ledge score, rotarod test performance and stride length when compared with controls [36].

#### 3.4.4. Retinal Degeneration

Two studies assessed retinal degeneration using the electroretinogram (ERG) [34,35]. Niu et al. proved that anti-ataxin-7 ASO injection before symptom onset led to increased cone and rod functions from 4 to 6 weeks post-treatment when compared with vehicle- or control ASO-treated mice [34]. However, when ASO treatment was administered after symptom onset, only the cone function was significantly better at 9 weeks [34]. ASOs targeting the CAG repeat led to no visual function improvement when compared to vehicle-treated subjects 6 weeks post-injection [34]. As for the ASO effect on retinal histology, when administered before symptom onset, there was an improvement in all retinal layer width compared with vehicle-treated mice; when administered after symptom onset, no improvement in retinal segments was observed [34]. Ramachandran et al. found no differences in neither mixed rod–cone response nor isolated cone response between iRNA anti-ataxin-7- and saline-treated mice at 30 weeks of age [35].

#### 3.4.5. Safety Profile

The safety profile was assessed by the two studies on iRNA [35,36]. Overall, both concluded that there was no toxicity associated with iRNA delivery. Anti-ataxin-7 iRNA subretinal injection led to no differences in retinal width or gliosis when compared to saline-injected eyes nor in retinal function, rod–cone response, isolated cone response and optokinetic tracking response [35]. In addition, intrathecal administration led to no astrocytic nor microglial activation when compared to WT mice [36].

### 3.5. Other HCAs

This review includes one paper on each of the following ataxias: spinocerebellar ataxia type 2 (SCA2), spinocerebellar ataxia type 6 (SCA6), Friedreich ataxia (FRDA) and ataxia–telangiectasia (A-T) [37,38,39,40].

Scoles et al. analysed the administration of ASOs in two SCA2 mouse models: one containing 127 CAG (ATAXN2-Q127) and another, closer to the human phenotype, containing 72 CAG (BAC-Q72) [40]. Regardless of the mouse model, ASO intrathecal injection at symptom onset suppressed ataxin-2 transcript levels. Regarding the neuropathological features, both models showed a near-complete restoration of normal Purkinje cell firing frequency [40]. Regarding motor behaviour, both showed better performance in the accelerating rotarod test than the SCA2 saline-treated group. Nonetheless, the performance was still far from that of WT mice. Concerning the safety profile, there was no significant astrocytic or microglial activation [40].

The *CACNA1A* gene encodes both the α1A protein (pore-forming protein) and the α1ACT protein (transcription factor involved in cerebellar development) [39]. A polyQ-expanded α1ACT protein is responsible for SCA6, with the α1A protein being essential to life. Therefore, the selective silencing of α1ACT would be the desirable approach, in contrast to full *CACNA1A* gene silencing [39]. On the basis that α1ACT is translated with an internal ribosome entry site (IRES), in contrast to α1A cap-dependent translation, Pastor et al. studied an miRNA that silenced these sequences within the *CACNA1A* IRES region. The authors tested an miRNA in a hyperacute SCA6 mouse model [39]. Regarding suppression efficacy, the miRNA-treated group showed lower mutant α1ACT protein levels than the control while keeping α1A levels similar to those of the control. When assessing neuropathological features, the authors identified a protective role in cerebellar molecular layer thinning, decreased dendritic tree density and a decreased number of Purkinje cells. In addition, motor deficits were improved in the treated group when compared with the control; better performance in the rotarod test, greater distances in the open-field assay and improved gait instability in all four limbs were observed [39].

Regarding FRDA, we identified one study on the CRISPR-Cas9 system [38]. The authors demonstrated its suppression efficacy *in vivo*, verifying GAA repeat excision with electroporation into the tibialis anterior muscle of an FRDA mouse model.

The included study on A-T demonstrated that the systemic administration of antisense morpholine oligonucleotides conjugated with arginine-rich cell penetrating peptides (AMOs-CPPs) in mice reached satisfactory delivery to the brain and Purkinje cells [37]. Still, the authors highlighted the need of studies focused on off-target effects and immune responses associated with AMOs-CPPs.

**Table 4 cells-12-01037-t004:** Key conclusions from data extraction sheet—other SCAs [37,38,39,40].

Authors	Year	HCA	GST	Key Conclusions
Scoles, D.R., et al.	2017	SCA2	ASOs	The intracerebral injection of ASO led to reduced ataxin-2 transcript and protein levels, resulting in delayed onset of SCA2 motor and neuropathological phenotypes without microglial activation.
Pastor, P.D.H., et al.	2018	SCA6	iRNA	iRNA selectively inhibited alpha-1ACT mutant protein and kept alpha-1A normal levels. It also prevented the development of motor and morphological abnormalities.
Ouellet, D.L., et al.	2016	FRDA	CRISPr/Cas9	CRISPR-Cas9 technology resulted in the excision of the GAA repeat in intron 1 using electroporation.
Du, L., et al.	2011	A-T	ASOs	Splicing correction efficiency of AMOs conjugated with CPPs was demonstrated *in vitro. In vivo*, systemic administration revealed efficient brain uptake, particularly in PC, without apparent signs of toxicity.

## 4. Discussion

As a result of the *a priori* established search and selection process, this review only included studies on MJD/SCA3, SCA7, SCA1, SCA6, SCA2, FRDA and A-T. No studies assessing suppression efficacy or neuropathological features in A-T, and motor behaviour or safety profile in A-T and FRDA were identified. Further investigations are needed in this field.

As a major limitation of this review, we acknowledge that study design specifications such as animal/disease model, GST dosage and timepoint evaluations were only briefly analysed. To overcome this limitation, a meta-analysis would have to be performed, but this was beyond the scope of the present work. Therefore, more robust conclusions about the most effective type of GST, optimal dosage, timepoint administration and evaluation cannot be drawn.

Outcome measures were identical across studies, harmonizing their analysis and description. The timepoint evaluation of outcomes varied according to the disease/animal models, and additional investigations would still be needed to define the best timeframes. Nonetheless, in animal and human studies, it is always crucial to evaluate short- and long-term effects on either suppression or phenotype efficacy as well as safety profile.

All selected studies proved to be effective in suppressing the targeted genes *in vivo*, even though they reported variable degrees of silencing magnitude. When assessed, GSTs were effective in improving neuropathological features, including decreasing neuronal inclusions, neuronal dysfunction and neuronal loss. With respect to the motor phenotype, every study, except one, proved that GSTs halted and improved motor performance in the intervention group [19,23,25,26]. Indeed, in this one exception, the authors found iRNA treatment to be effective in suppressing the targeted gene, as well as improving neuropathological features [20]. Thus, they concluded that either the administration timing was too late to revert symptoms and/or pathogenesis was not limited to the cerebellum [20].

Concerning treatment timing, GSTs administered either before or after symptom onset proved to be effective in improving motor performance in MJD/SCA3 and SCA1, and eye degeneration in SCA7 [19,23,25,26,28,33]. If administered before symptom onset or in early symptomatic mice, motor performance was similar to that of WT mice, and eye degeneration was more attenuated than in the groups treated after symptom onset. Even though most studies suggested that the earlier the administration is, the more effective the treatment is, the lack of studies on late-stage models could be biasing this conclusion.

Only two studies on iRNA in MJD/SCA3 mice focused on the WT protein role, suggesting no apparent benefit to specifically silencing the mutant allele or WT protein overexpression [17,18]. Nonetheless, there exists the possibility that WT protein at higher levels than the ones tested could be beneficial. Thus, further investigations are still needed, including investigations of effects beyond neuropathological features, such as the motor phenotype.

The most commonly identified adverse effects included glial and astrocytic activation. This non-neuronal response, when present, was related to the injection, regardless of it being GST or placebo. Indeed, most studies reported intrathecal administration leading to glial and astrocytic activation limited to the injection site and resolving over time. These findings suggest that GSTs are safe. Intrathecal injections are a direct route for delivering the drug, surpassing the blood–brain barrier and significantly decreasing the risk of liver/renal metabolism, drug interactions and systemic toxicity. Still, knowledge on central nervous system toxicity is still limited, and more investigations are needed. Peripheral administrations, which are more convenient for patients, have the systemic metabolism inconvenient but may be promising and are worthy of further studies [19,40].

Interestingly, Vásquez-Mojena et al., in a recent detailed narrative review of in vitro and *in vivo* GST studies on HCAs, also found promising results of iRNA and ASOs [41]. Additionally, the authors highlighted the limited use of CRISPR/Cas9 thus far. Even though the adopted methodology for the reviews was different, we share the main conclusions.

We believe that one of the most important strengths of our work was assessing the risk of bias. After a detailed analysis, we verified that most studies were not clear on the parameters used, with no guarantees of results and conclusions being free of bias. This issue has already been addressed for animal studies, and ARRIVE guidelines have been created to overcome it [42]. Still, its use is far behind what would be desirable. It is our opinion that randomization and blinding should be routinely implemented in these studies, standardizing this crucial part of medical investigations, making them similar to human clinical trials. The impact of bias on research on animal models is worthy of further investigations.

With this review, we wished to reunite the information available on GSTs in HCA animal models and present it in a clear way for clinicians. The first human studies are already on their way, and this could serve as a basis to better understand the knowledge available thus far.

## Figures and Tables

**Figure 1 cells-12-01037-f001:**
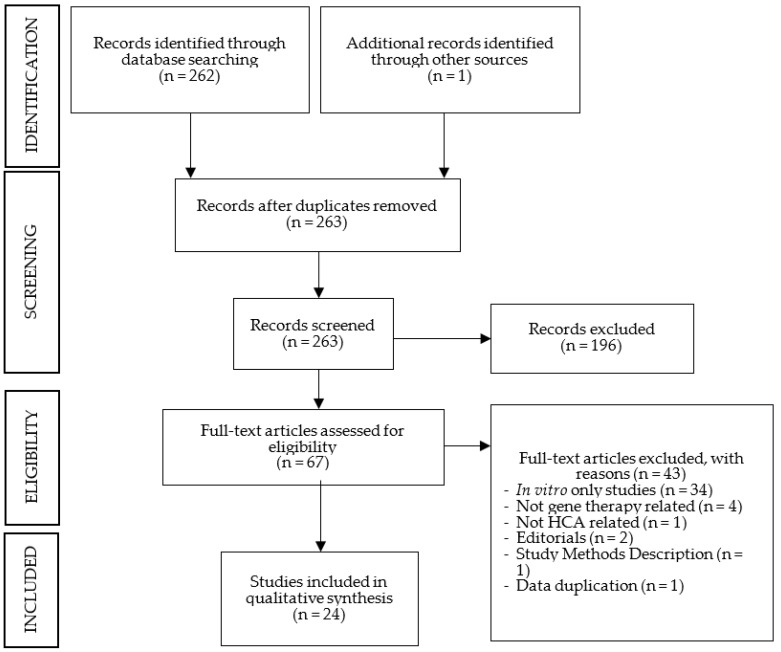
Search results in a flow diagram (adapted from PRISMA 2009).

**Figure 2 cells-12-01037-f002:**
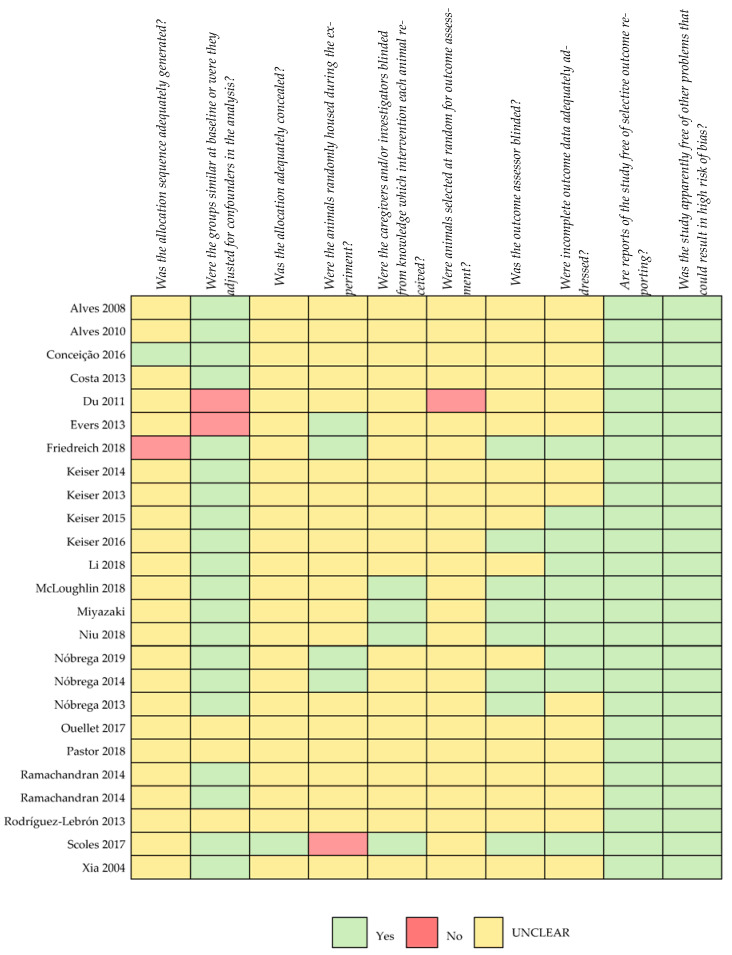
Risk-of-bias assessment (adapted from SCYRCLE’s risk-of-bias tool) [17,18,19,20,21,22,23,24,25,26,27,28,29,30,31,32,33,34,35,36,37,38,39,40].

## Data Availability

Research data have not been archived at a public repository but are available upon request from CHUPorto.

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
