# Peer review of "Gene Suppression Therapies in Hereditary Cerebellar Ataxias: A Systematic Review of Animal Studies"

_cells, 2023, doi:10.3390/cells12071037_

Round 1

Reviewer 1 Report

An appreciated review of current status of evidence on GST in hereditary ataxias from animal studies. I specifically appreciated the clear structuring of content, presented per genotype and per domain of effects and presenting risk of bias assessment. The description of literature search is sound and may remain as described, but should discuss expectedly resulted in some redundancy. Confinement of search towards original work/ in vivo/ animal studies could have possibly excluded large part of the records which here have been excluded based on heading and abstract. More importantly, differences in animal models and modes of GST application/ location/ dosing as well as observation schedule should be addressed in results section and implications for the interpretation and comparison of efficacy/safety data should be discussed. For example, the issue of GST effects versus disease stage seems somewhat hidden in text and could be presented more clearly. Further, I suggest to summarize either in intro or as part of results the steps applied in original publications reviewed to prove efficacy and safety of GST in animal studies, this would be useful guidance for the reader to follow per genotype paras. Authors may aim to relate reviewed results to more general considerations on this topic in discussion: which of the steps are considered useful and sufficient for this purpose, e.g. is there guidance on appropriate timeframes for observation to investigate efficacy/safety measures for the animal models applied? Related to this point, authors may consider to include information on animal model/disease stage and dosing scheme in tables 1 - 4.  Discussion mentions bias as a prominent point. This should be extended towards suggestions for improvement/ reduction of bias, possibly relating to reference of relevant published guidelines. Related to this aspect: to what extent can non-blinding of investigators be expected to bias results? how many of the findings reported do the authors consider replicated in an independent study? Discussion of efficacy results should reflect comparability among studies in assessment and timeframes of observation. Conclusion on lower efficacy in longer standing disease – though plausible – may also be biased by the fact that most records relate to pre-symptomatic/ early-symptomatic animals. Further, authors may help the reader understand how reviewed records may guide the way for future study in humans, e.g. timepoint and mode of application, safety caveats, developments for single genetic entities, factors of preference for systems of GST reported herein.

Some minor points in editing: figure S1 referenced in text page 3 line 109 seems to be identical to figure 2 presented on next page? Insert in text 3.2.2. on page 5, line 135-138 seem misplaced and should possibly be deleted. With respect to tissue layers wording of thicker instead of larger/wider seems more appropriate. Vitreous body instead of vitreous humor. Wording page 6, line 220 (translating) should be checked, should this mean which implies? For FRDA (page 11) add intronic GAA triplet expansion.

As a last important point, unfortunately, numbering of references got obviously mixed up and needs careful check before re-submission. Suggestion is, to also check and report, if separate records were published form the same lab, e.g. both records on SCA6 are cited in parallel, do they refer to different publications form the same experiment? 

Author Response

1- An appreciated review of current status of evidence on GST in hereditary ataxias from animal studies. I specifically appreciated the clear structuring of content, presented per genotype and per domain of effects and presenting risk of bias assessment. The description of literature search is sound and may remain as described, but should discuss expectedly resulted in some redundancy. Confinement of search towards original work/ in vivo/ animal studies could have possibly excluded large part of the records which here have been excluded based on heading and abstract.

Author response: We are much grateful to the Reviewer for the attentive comments. Discussion was re-written as to avoid redundancy.  We agree with the reviewer that as our inclusion criteria considered only animal studies, all in vitro studies were not assessed. As one of our main objectives was to assess risk of bias, we excluded in vitro studies. As to in vivo studies, the literature search was performed by two independent individuals, in all steps of the process, decreasing the risk of excluding important studies. For sure this cannot be 100% ruled out, but our methodology was in accordance with the Prisma guidelines.

2- Differences in animal models and modes of GST application/ location/ dosing as well as observation schedule should be addressed in results section and implications for the interpretation and comparison of efficacy/safety data should be discussed. For example, the issue of GST effects versus disease stage seems somewhat hidden in text and could be presented more clearly.

Author response: We appreciate the reviewer’s comment. We have tried to always highlight the time in disease course where the GST were applied, particularly when describing motor behaviour, and have re-written these parts (e.g. pages 7, 9-11). Still, to be able to draw more robust conclusions, a meta-analysis would have to be performed, but was beyond the scope of this work. We have added a limitations paragraph on discussion, to make this clearer.

3 - Further, I suggest to summarize either in intro or as part of results the steps applied in original publications reviewed to prove efficacy and safety of GST in animal studies, this would be useful guidance for the reader to follow per genotype paras.

Author response: Again, we thank the reviewer for this pertinent suggestion, and have added this information (first paragraph, page 6).

4 - Authors may aim to relate reviewed results to more general considerations on this topic in discussion: which of the steps are considered useful and sufficient for this purpose, e.g. is there guidance on appropriate timeframes for observation to investigate efficacy/safety measures for the animal models applied?

Author response: Thank you for the suggestion, we have added this information in discussion.

5 - Related to this point, authors may consider to include information on animal model/disease stage and dosing scheme in tables 1 - 4. 

Author response: As to time restrictions we were not able to include this information in the tables. Still, we have highlighted it in the text, and acknowledge that a meta-analysis would be the preferred methodology for it. Hope the reviewer considers this enough.

6 - Discussion mentions bias as a prominent point. This should be extended towards suggestions for improvement/ reduction of bias, possibly relating to reference of relevant published guidelines.

Author response: Again, we would like to thank the reviewer for the pertinent comment. In fact risk of bias assessment to be the major strength of our work. This information was added on discussion

7 - Related to this aspect: to what extent can non-blinding of investigators be expected to bias results?

Author response: We believe that, as in humans’ clinical trials, blinding would be important to reduce bias. Everyone who is conducting research, particularly in disease modifying strategies of such debilitating disorders, as an unconscious (and possibly also conscious) will to see positive results. As opposed to humans, we believe the placebo effect in the control group would not be a problem in animal studies, but still it would have to be studied! We have added information on this topic on discussion.

8 - Discussion of efficacy results should reflect comparability among studies in assessment and timeframes of observation.

Author response: As we have not performed a meta-analysis, comparisons among studies and timeframes of observations would be difficult to perform. As so, we have reflected the general conclusions of all trials and again, acknowledge our limitations in discussion.

9 - Conclusion on lower efficacy in longer standing disease – though plausible – may also be biased by the fact that most records relate to pre-symptomatic/ early-symptomatic animals.

Author response: We completely agree with the reviewer, and have added this information in discussion.

10 - Further, authors may help the reader understand how reviewed records may guide the way for future study in humans, e.g. timepoint and mode of application, safety caveats, developments for single genetic entities, factors of preference for systems of GST reported herein.

Author response: We are grateful for the suggestion and have added a paragraph on discussion.

11 - Some minor points in editing:

  • Figure S1 referenced in text page 3 line 109 seems to be identical to figure 2 presented on next page?
  • Insert in text 3.2.2. on page 5,
  • Line 135-138 seem misplaced and should possibly be deleted.
  • With respect to tissue layers wording of thicker instead of larger/wider seems more appropriate.
  • Vitreous body instead of vitreous humor.
  • Wording page 6, line 220 (translating) should be checked, should this mean which implies?
  • For FRDA (page 11) add intronic GAA triplet expansion.

Author response: We are truly sorry for these typos, and have corrected it al.

12 - As a last important point, unfortunately, numbering of references got obviously mixed up and needs careful check before re-submission. Suggestion is, to also check and report, if separate records were published form the same lab, e.g. both records on SCA6 are cited in parallel, do they refer to different publications form the same experiment? 

Author response: Again, we’re sorry for this. We had a problem with our references’ software. We have deleted it all, and reorganized all our references.

Reviewer 2 Report

The manuscript by Santos and collaborators aims at summarizing the state of the art about gene suppression therapy in hereditary cerebellar ataxias. They reviewed papers reporting studies on animal models (excluding those limited to in vitro/ex vivo experiments), published in english/portuguese.
The result is a compendium of up-to-date literature that can be useful for researchers as a practical reference. At the same time, it highlights the potential pitfalls in performing meta-analysis/systematic reviews due to the methodological flaws of some published papers.
The cited literature seems complete, in accordance with the established criteria, and results are reported in a clear and concise way.

Major comments:

- There is a recent 2021 paper by Vázquez-Mojena et al. :"Gene Therapy for Polyglutamine Spinocerebellar Ataxias: Advances, Challenges, and Perspectives", PMID: 34628681, in which several of the studies reported by Santos and collaborators are discussed (in vitro/ex vivo studies are also reported). This work should be cited and the similarities/differences in the conclusion drawn should be discussed.

- Material and Methods: a brief sentence could be added explaining why in vitro/ex vivo studies have been excluded.

- Lines 194-201 and discussion: The conclusions about the role of the WT protein should be better explained.

Minor comments:
Abstract, line 29: It is better to cite SCAs in numerical order.
Abstract, line 33: "Cas9" has no space before the number.
Introduction: Ataxia-telangiectasia, reported in results, should be mentioned.
Search outcome and Figure 1: In the text "41 articles were excluded, and an additional article was included" but in the figure "Full-text articles excluded, with reasons (n = 42)".
Also, one of the label is missing a letter (ELIGIBILIT -> ELIGIBILITY)
Figure 2 is not cited in the text. There is also a question mark missing from the label of the fist column.
Lines 136-138 should be deleted.
Table 1-4: the "AUTHOR" columns could be replaced by a numerical reference, leaving more space for the rest of the text.
Paragraph 3.5 Other HCAs: Why did the author choose the order SCA5-FRDA-SCA2-AT? A more straightforward approach could have been dominant forms - SCA2, SCA5; recessive forms - FRDA, AT.
Line 339: Reference 38 should be PMID: 30922876 by Du, Wei, Pastor et al.
Line 367: Reference 36 should point to the current reference 38 PMID: 21576124 by Du, Kayali et al. Please double check all the other references.
Line 413: Since ex vivo/in vitro studies have been excluded by design, the results do not report "all the information on GST for HCA" and this should be highlighted.

Author Response

The manuscript by Santos and collaborators aims at summarizing the state of the art about gene suppression therapy in hereditary cerebellar ataxias. They reviewed papers reporting studies on animal models (excluding those limited to in vitro/ex vivo experiments), published in english/ portuguese.
The result is a compendium of up-to-date literature that can be useful for researchers as a practical reference. At the same time, it highlights the potential pitfalls in performing meta-analysis/systematic reviews due to the methodological flaws of some published papers.
The cited literature seems complete, in accordance with the established criteria, and results are reported in a clear and concise way.

Author response: We are much grateful to the Reviewer for the kind comments.

Major comments:

1. There is a recent 2021 paper by Vázquez-Mojena et al. :"Gene Therapy for Polyglutamine Spinocerebellar Ataxias: Advances, Challenges, and Perspectives", PMID: 34628681, in which several of the studies reported by Santos and collaborators are discussed (in vitro/ex vivo studies are also reported). This work should be cited and the similarities/differences in the conclusion drawn should be discussed.

Author response: We are very grateful to the reviewer for this excellent suggestion. The paper by Vásquez-Mojena is a wonderful review and we have added this information on discussion. Even though the methodology used was different, we believe the main conclusions are similar. Just of note, the importance of bias in animal studies should be a matter of concern for all the community. We wonder if that could be one of the reasons for clinical trials having less success than animal studies.

2.  Material and Methods: a brief sentence could be added explaining why in vitro/ex vivo studies have been excluded.

Author response: We have added this information

  1. Lines 194-201 and discussion: The conclusions about the role of the WT protein should be better explained.

Author response: We are grateful for the comment. WE have added information on the role of WT in introduction, also on results and discussion.

Minor comments:
- Abstract, line 29: It is better to cite SCAs in numerical order.

Author response: We have updated as requested

- Abstract, line 33: "Cas9" has no space before the number.

Author response: We have updated as requested

- Introduction: Ataxia-telangiectasia, reported in results, should be mentioned.

Author response: We have updated as requested

- Search outcome and Figure 1: In the text "41 articles were excluded, and an additional article was included" but in the figure "Full-text articles excluded, with reasons (n = 42)".

Author response: We are truly grateful for identifying this typo and have updated.

- Also, one of the label is missing a letter (ELIGIBILIT -> ELIGIBILITY)

Author response: We have updated as requested

- Figure 2 is not cited in the text. There is also a question mark missing from the label of the fist column.

Author response: We have updated as requested

- Lines 136-138 should be deleted.

Author response: We have updated as requested

- 1-4: the "AUTHOR" columns could be replaced by a numerical reference, leaving more space for the rest of the text.

Author response: We have reduced the size of the font, to keep the author’s name. We thought this way the tables would be easier to read, and would like to ask for the reviewer license to keep this way.

- Paragraph 3.5 Other HCAs: Why did the author choose the order SCA5-FRDA-SCA2-AT? A more straightforward approach could have been dominant forms - SCA2, SCA5; recessive forms - FRDA, AT.

Author response: We have updated as requested

- Line 339: Reference 38 should be PMID: 30922876 by Du, Wei, Pastor et al.
- Line 367: Reference 36 should point to the current reference 38 PMID: 21576124 by Du, Kayali et al. Please double check all the other references.

Author response: We’re truly sorry for this. We had a problem with our references’ software. We have deleted it all, and reorganized all references.

- Line 413: Since ex vivo/in vitro studies have been excluded by design, the results do not report "all the information on GST for HCA" and this should be highlighted.

Author response: We completely agree with the reviewer and have changed the text.

Reviewer 3 Report

In this manuscript, the authors present a review on in vivo preclinical studies investigating the efficacy and safety profile of gene suppression therapies (based on RNAi and ASO) in rodent models of hereditary cerebellar ataxias. From a large literature search, they selected 25 critical studies for in-depth reporting; all but two relate to polyglutamine spinocerebellar ataxias.

The review is mainly a description of the main outcomes of each study, without taking into account the severity of the phenotypes of the different animal models treated or the different methods of administration of the reagents (systemic, intracerebellar, intrathecal, vectorized or non-vectorized reagents, etc). Relevantly, the authors point out the lack of clarity in the risk of bias assessment of these studies. However, there is very little discussion of strengths and other limitations of these studies, or lessons that can be learned from this review that could guide new preclinical development of gene suppression therapies. For these reasons, the value of this review remains modest.

I have a few recommendations and minor comments:

1.     The references need to be reformatted. This is an undergraduate type of mistake, that made the assessment of the manuscript by this referee quite confusing. e.g. Ref 2 and 3 are the same, Ref 9 and 10 are the same, etc.

2.     Why the 4 tables of the main manuscript are also presented, as supplementary tables (Table-SII to table-SV).

3.     The listing in each table does not seem to follow any logic, as by tool, by year of publication, by team.

4.     In tables, the author first name is not useful; only the author family name is required to match with the main text. e.g. Maria do Carmo in the table 1 does not match Costa et al in lanes 127, 183, 204 and 210 or in Figure 2. To match the text, reference list and table, add the reference number in the table as it appears in the reference list.

5.     Lane 46: Friedreich ataxia instead of Friederich ataxia.

6.     Lane 46: Friedreich ataxia is not a polyQ disorders.

7.     Lane 48: Although the wild-type (WT) protein role is not fully understood, it has been suggested to have a protective effect.[5,6] This applies for WT ataxin-3 only and cannot be used a general statement.

8.     Lane 62: siRNA can also lead to translation inhibition.

9.     Lane 68: The authors may want to finish this section by describing the different therapeutic impact of targeting RNA (RNAi, ASO) versus Crispr/Cas9 (DNA).

10.  Lane 109: is figure S1 the figure 2 of this main manuscript?

11.  Lane 112: this figure is excellent!

12.  Lane 132: the reference Li et al about knockdown efficiency in fly does not seem relevant here.

13.  Lane 135-137: ?????

14.  Lane 158: DARPP32 is a striatal neuron marker.

15.  Lane 170: thickness of ML??? values?? Are they micrometers?

16.  Lane 192: in-text reference formatting.

17.  Lane 266 : the term “control” is often unclear as to refer to “untreated mutant mice”,  saline or vehicle treated mice” or “WT mice”. This should be better specified here and also elsewhere along the text.

18.  Lane 300: the use of different nomenclature creates confusion, as for antiataxin-7 iRNA or lane 303 iRNA anti ataxin-7.

19.  Lane 320-323: the reported data have very little significance. The SCA7 mouse model used in ref 34 did not have any retina dysfunction; therefore, their results only showed that RNAi treatment did not cause per se any adverse effect on retinal function in these mice. This should be made clearer in the text or delete lane 320-323.

20.  Lane 351-355: why FRDA treatment is not listed in table 4?

21.  Discussion: The authors may wish to discuss the future direction of gene suppression therapy. In particular, progress has been made in the development of crispr/Cas9 technology to suppress repeated CAG expansion in Huntington's disease, which promises a new development for hereditary cerebellar ataxias. Another important point of discussion is whether allele-specific targeting would have better safety outcome in hereditary cerebellar ataxia, given that treatment in patients will last decades.

Author Response

In this manuscript, the authors present a review on in vivo preclinical studies investigating the efficacy and safety profile of gene suppression therapies (based on RNAi and ASO) in rodent models of hereditary cerebellar ataxias. From a large literature search, they selected 25 critical studies for in-depth reporting; all but two relate to polyglutamine spinocerebellar ataxias.

The review is mainly a description of the main outcomes of each study, without taking into account the severity of the phenotypes of the different animal models treated or the different methods of administration of the reagents (systemic, intracerebellar, intrathecal, vectorized or non-vectorized reagents, etc). Relevantly, the authors point out the lack of clarity in the risk of bias assessment of these studies. However, there is very little discussion of strengths and other
limitations of these studies, or lessons that can be learned from this review that could guide new preclinical development of gene suppression therapies. For these reasons, the value of this review remains modest.

Author response: We have tried to summarize all the relevant information from each study, and present it in a clear and organized way for clinicians. The timepoint of administration as well as the type of administration were also taken into account (mostly intrathecal). A meta-analysis would have to be performed in order to have that information processed in more detail, but was not on the scope of this work

I have a few recommendations and minor comments:

1.     The references need to be reformatted. This is an undergraduate type of mistake, that made the assessment of the manuscript by this referee quite confusing. e.g. Ref 2 and 3 are the same, Ref 9 and 10 are the same, etc.

Author response: We’re truly sorry for this. We had a problem with our references’ software. We have deleted it all, and reorganized all references.

  1.    Why the 4 tables of the main manuscript are also presented, as supplementary tables (Table-SII to table-SV).

Author response: We have corrected this typo

  1.    The listing in each table does not seem to follow any logic, as by tool, by year of publication, by team.

Author response: We have reorganized the tables

  1.    In tables, the author first name is not useful; only the author family name is required to match with the main text. e.g. Maria do Carmo in the table 1 does not match Costa et al in lanes 127, 183, 204 and 210 or in Figure 2. To match the text, reference list and table, add the reference number in the table as it appears in the reference list.

Author response: We have reorganized the information

  1.    Lane 46: Friedreich ataxia instead of Friederich ataxia.

Author response: We have reorganized the typo

  1.    Lane 46: Friedreich ataxia is not a polyQ disorders.

Author response: We completely agree with the reviewer and have corrected this fault.

  1.    Lane 48: Although the wild-type (WT) protein role is not fully understood, it has been suggested to have a protective effect. [5,6] This applies for WT ataxin-3 only and cannot be used a general statement.

Author response: We have rephrased this part of the introduction.

  1.    Lane 62: siRNA can also lead to translation inhibition.

Author response: We have rephrased it.

9.  Lane 109: is figure S1 the figure 2 of this main manuscript?

Author response: We have corrected this typo.

10.  Lane 112: this figure is excellent!

Author response: We are grateful for the kind comment.

11.  Lane 132: the reference Li et al about knockdown efficiency in fly does not seem relevant here.

Author response: We have excluded this reference from the neuropathology section.

12.  Lane 135-137: ?????

Author response: We thank the reviewer for identifying this mistake and have rephrased it

13.  Lane 158: DARPP32 is a striatal neuron marker.

Author response: We have detailed this information

14.  Lane 170: thickness of ML??? values?? Are they micrometers?

Author response: We have updated this information

15.  Lane 192: in-text reference formatting.

Author response: We have updated this information

16.  Lane 266 : the term “control” is often unclear as to refer to “untreated mutant mice”,  saline or vehicle treated mice” or “WT mice”. This should be better specified here and also elsewhere along the text.

Author response: We have updated this information

17.  Lane 300: the use of different nomenclature creates confusion, as for antiataxin-7 iRNA or lane 303 iRNA anti ataxin-7.

Author response: We agree with standardizing designations, and apologize for not having performed it previously

18.  Lane 320-323: the reported data have very little significance. The SCA7 mouse model used in ref 34 did not have any retina dysfunction; therefore, their results only showed that RNAi treatment did not cause per se any adverse effect on retinal function in these mice. This should
be made clearer in the text or delete lane 320-323.

Author response: We are thankful to the reviewer for this comment and have updated the information.

  1. Lane 351-355: why FRDA treatment is not listed in table 4?

Author response: We are thankful to the reviewer for noting this typo and have updated the information.

  1. Discussion: The authors may wish to discuss the future direction of gene suppression therapy. In particular, progress has been made in the development of crispr/Cas9 technology to suppress repeated CAG expansion in Huntington's disease, which promises a new development for hereditary cerebellar ataxias. Another important point of discussion is whether
    allele-specific targeting would have better safety outcome in hereditary cerebellar ataxia, given that treatment in patients will last decades.

Author response: We have added this information